

# Introducing iFluid: a numerical framework for solving hydrodynamical equations in integrable models

**Frederik S. Møller[*] and Jörg Schmiedmayer**

Vienna Center for Quantum Science and Technology, Atominstitut,
TU Wien, Stadionallee 2, 1020 Vienna, Austria

* frederik.moller@tuwien.ac.at

## Abstract

We present an open-source Matlab framework, titled iFluid, for simulating the dynamics of integrable models using the theory of generalized hydrodynamics (GHD). The framework provides an intuitive interface, enabling users to define and solve problems in a few lines of code. Moreover, iFluid can be extended to encompass any integrable model, and the algorithms for solving the GHD equations can be fully customized. We demonstrate how to use iFluid by solving the dynamics of three distinct systems: (i) The quantum Newton's cradle of the Lieb-Liniger model, (ii) a gradual field release in the XXZ-chain, and (iii) a partitioning protocol in the relativistic sinh-Gordon model.



# 1  Introduction

In recent decades great experimental advances in the field of ultracold atoms have enable the preparation and manipulation of one-dimensional many-body quantum system far from equilibrium [1–9]. Therefore, theoretical tools for understanding the complex dynamics of these systems have been highly sought after [10–12]. Some of these low dimensional systems exhibit integrability by abiding to an infinite set of local conservation laws [13]. For a long time, integrable models have been a theoretical playground, although several of these models have also been realized experimentally [1,6,7,9].

Recently the theory of generalized hydrodynamics (GHD) has emerged as a powerful framework for studying integrable models out of equilibrium [14,15]. In its most basic form, generalized hydrodynamics describes the flow of all the conserved charges of integrable models. Thus, an infinite set of advection equations emerge, which through the thermodynamic Bethe ansatz, can be formulated as a single Euler-scale equation for a quasiparticle distribution. Since the inception of GHD several applications have been added to the framework, such as calculations of entanglement spreading [16–19], correlation functions [20–22], diffusive corrections [23,24], and many others [25–28]. Recently it has also been demonstrated to capture the dynamics of a cold Bose gas trapped on an atom-chip [29]. An especially appealing feature of the GHD framework is how the main equations can be universally applied to a large set of integrable models including the Lieb-Liniger model [14,30–32], XXZ chain [15,33–35], classical [36] and relativistic [14] sinh-Gordon, and many more [37–40]. The theory has already proven its worth by providing exact predictions for the many-body dynamics in several cases [14,15,20]. Additionally, GHD appears to have great potential as a numerical tool, as the computational complexity of solving the many-body dynamics is entirely independent of the Hilbert space size. Despite this, only a couple of different numerical schemes have been implemented so far [27,28,34,38,41]. Thus, if GHD is to applied on larger scales, such as describing experimental observations, more powerful numerical methods must be developed.

Currently, no open-source code exists for solving GHD equations. The goal of iFluid (integrable-Fluid) is to provide a powerful and intuitive numerical framework for finding and propagating the root density distribution, which serves as the basic quantity for thermodynamic calculations in integrable systems [42]. Hence, iFluid supplies a platform for solving the core hydrodynamical equations on top of which user-specific applications can be built. The universality of the GHD equations enables a highly flexible code base, wherein any integrable model can be seamlessly integrated. iFluid already supports a couple of models (see Appendix A), and the implementation of a new model can be achieved in relatively few lines of code after extending core classes of the framework.

So far only little effort has been put into comparing different algorithms for solving the GHD equations. For very cold temperatures the underlying quasiparticle distribution resembles that of a Fermi sea, whose hard walls and edges complicates numerical solutions. However, such problems have already been studied for many years within the field of fluid dynamics [43]. Adopting these methods should greatly bolster the numerical capability of generalized hydrodynamics. Therefore, iFluid abstracts the algorithm for solving the main advection equation,

whereby users are free to implement whatever algorithm is suitable for their specific problem. Again, iFluid already implements a couple of basic algorithms sufficient for solving most tasks.

The purpose of this paper is to introduce the user to the iFluid framework and demonstrate its applicability in various scenarios. Thus, the paper is organized as follows. In Section 2, we review the basic concepts of GHD serving as the core of the iFluid framework. In section 3, the core features of iFluid are discussed. In Section 4, the applicability of iFluid is demonstrated by solving three distinct problems: The quantum Newtons cradle in the Lieb-Liniger model, a gradual confinement release in the XXZ model, and a partitioning protocol in the relativistic sinh-Gordon model. Finally in Section 5, we conclude and give an give an outlook for the development of iFluid. Details of the exact numerical implementations are reported in appendices.

## 2 Review of generalized hydrodynamics (GHD)

Generalized hydrodynamics in its essence utilizes the quasiparticle formulation of the thermodynamics Bethe Ansatz to describe the flow of charges within an integrable system. Integrable systems abide to infinitely many local conservation laws [44], thus preventing a conventional hydrodynamical description which only captures conservation of particles, momentum, and energy. This infinite set of conserved charges imposes constraints on the dynamics of the system and inhibits thermalization. Hence, under the assumption of local thermal equilibrium, the systems relaxes to a generalized Gibbs ensemble (GGE) from which thermodynamic quantities can be derived [45]. Once it is at this stage, the system can be described via the quasi-particles based thermodynamic Bethe ansatz (TBA). Within TBA the eigenstates of the full set of local conserved charges are multiparticle states, with each particle labelled by a rapidity $\lambda$ [13]. Under integrability all multiparticle scattering events factorize into two-body elastic scatterings. Thus, all interactions between the quasiparticles are captured by the two-body scattering matrix $S(\lambda)$. In the thermodynamic limit the rapidity can be thought of as a continuous variable, while the coarse-grained root density $\rho(\lambda)$ gives the density of particles within the interval $[\lambda, \lambda + d\lambda]$ [42]. The root densities (like the GGE density matrix, $\hat{\rho}_{\text{GGE}}$) fix the expectation value of the the local charges, $\hat{Q}_j$, such that [46]

$$\frac{1}{L}\langle \hat{Q}_j \rangle = \frac{1}{L}\text{Tr}\big[\hat{Q}_j \hat{\rho}_{\text{GGE}}\big] = \int d\lambda \, h_j(\lambda)\rho(\lambda) \equiv q_j \,, \tag{1}$$

where $h_j(\lambda)$ is called the one-particle eigenvalue of the charge $\hat{Q}_j$, and $L$ is the system length. Among the infinite set of conserved charges we find the particle number $\hat{Q}_0 = \hat{N}$, the total momentum $\hat{Q}_1 = \hat{P}$, and the total energy $\hat{Q}_2 = \hat{E}$. Thus, the corresponding one-particle eigenvalues are $h_0(\lambda) = 1$, $h_1(\lambda) = p(\lambda)$, and $h_2(\lambda) = \epsilon(\lambda)$, where $p(\lambda)$ and $\epsilon(\lambda)$ are the momentum and energy of a single quasiparticle respectively.

In a similar fashion we can calculate the expectation values of the currents associated to the charges via

$$j_j \equiv \int d\lambda \, h_j(\lambda) v^{\text{eff}}(\lambda)\rho(\lambda) \,, \tag{2}$$

where $v^{\text{eff}}(\lambda)$ is the velocity by which the quasiparticles move. Later we will see how exactly this velocity is computed.

Until recently, the thermodynamic Bethe ansatz was used only to describe the expectation values of a homogeneous, stationary state. Imagine however, a weakly inhomogeneous system, where physical properties vary on space-time scales much larger than the underlying time

scales. In this case local equilibrium is established faster than the physical quantities can change, whereby the overall system remains in a quasi-stationary state. Thus, the system can be thought of as consisting of space-time fluid cells, each of which is described by a local GGE with minimal variations to those of neighbouring cells [14]. For a quasi-stationary state the quasiparticle description of the thermodynamic Bethe ansatz still applies, however, the root density is now weakly dependent on time and space, $\rho = \rho(t, x, \lambda)$. The main feature of GHD is formalizing how the root density behaves under time evolution. Thus, the GHD details how the flow of an infinite set of charges of an integrable model is given by the semiclassical propagation of a phase-space density of a quasiparticle collection.

In practice, it is more convenient to express the hydrodynamical equations in terms of the filling function, $\vartheta(\lambda)$. The filling can be interpreted as the density of quasiparticles over the density of available states at a given rapidity, $\rho_s(\lambda)$, such that

$$\vartheta(\lambda) = \frac{\rho(\lambda)}{\rho_s(\lambda)} = \frac{2\pi\rho(\lambda)}{(\partial_\lambda p)^{\mathrm{dr}}} \,. \tag{3}$$

Note that we have omitted the spacial and temporal argument for the sake of lighter notation. The subscript $^{\mathrm{dr}}$ in eq. (3) denotes the dressing of the quantity. Non-trivial interactions between the particles induces collective effects, which are captured by the dressing operation defined through the integral equation

$$h^{\mathrm{dr}}(\lambda) = h(\lambda) - \int \frac{\mathrm{d}\lambda'}{2\pi} \partial_\lambda \Theta(\lambda - \lambda') \vartheta(\lambda') h^{\mathrm{dr}}(\lambda') \,. \tag{4}$$

Due to the factorization of scatterings, the interparticle interactions are captured solely by the two-body scattering phase $\Theta(\lambda) = -i \log S(\lambda)$, with $S(\lambda)$ being the two-body scattering matrix. The interaction between particles also influences their equations of motion. In the non-interacting case the particles move with the group velocity, $v(\lambda) = \partial_\lambda \epsilon / \partial_\lambda p$. However, in the presence of interactions the bare quantities become dressed, whereby the particles move with an effective velocity [14, 15]

$$v^{\mathrm{eff}}(\lambda) = \frac{(\partial_\lambda \epsilon)^{\mathrm{dr}}}{(\partial_\lambda p)^{\mathrm{dr}}} \,. \tag{5}$$

Furthermore, space-time inhomogeneities in the parameters of the model induce force terms on the quasiparticles, which can change their rapidities. Once again the interparticle interactions collectively dress these force terms, whereby the quasiparticles experience an effective acceleration [27]

$$a^{\mathrm{eff}}(\lambda) = \frac{\partial_t \alpha f^{\mathrm{dr}} + \partial_x \alpha \Lambda^{\mathrm{dr}}}{(\partial_\lambda p)^{\mathrm{dr}}} \,, \tag{6}$$

where $\alpha$ is a coupling of the model (such as the interaction strength $c$ in the Lieb-Liniger model [30]). Inhomogeneities of the couplings in space and time have their own associated force term given respectively by

$$f(\lambda) = -\partial_\alpha p(\lambda) + \int \frac{\mathrm{d}\lambda'}{2\pi} \partial_\alpha \Theta(\lambda - \lambda') (\partial_\lambda p)^{\mathrm{dr}} \vartheta(\lambda') \tag{7}$$

and

$$\Lambda(\lambda) = -\partial_\alpha \epsilon(\lambda) + \int \frac{\mathrm{d}\lambda'}{2\pi} \partial_\alpha \Theta(\lambda - \lambda') (\partial_\lambda \epsilon)^{\mathrm{dr}} \vartheta(\lambda') \,. \tag{8}$$

Finally, the evolution of the phase-space quasiparticle density is captured with the single hydrodynamical equation [26, 27]

$$\partial_t \vartheta(\lambda) + v^{\mathrm{eff}}[\vartheta(\lambda)] \partial_x \vartheta(\lambda) + a^{\mathrm{eff}}[\vartheta(\lambda)] \partial_\lambda \vartheta(\lambda) = 0 \,, \tag{9}$$

where the brackets explicitly indicates that the velocity and acceleration is dependent on the current state. Eq. (9) is a simple Eulerian fluid equation, which describes the flow of the infinite set of conserved charges through a single expression. It is the main equation of GHD along with its root density-based counterpart

$$\partial_t \rho(\lambda) + \partial_x \left( v^{\text{eff}}[\rho(\lambda)] \rho(\lambda) \right) + \partial_\lambda \left( a^{\text{eff}}[\rho(\lambda)] \vartheta(\lambda) \right) = 0 \,, \tag{10}$$

which identifies the root density as a conserved fluid density [14]. From a numerical perspective, eq. (9) is more convenient to work with than eq. (10).

While the thermodynamic states of the Lieb-Liniger model are characterized by a single root density, this is in general not the case. For instance, the XXZ chain supports bound states, which are captured by including multiple types of quasi-particles, each with their own corresponding root density, $\rho^k(\lambda)$. Thus, one must sum over the contribution from each quasi-particle type to obtain the charge densities as described in eq. (1). To simplify the notation we adopt the convention from [26] of writing the rapidity as a single spectral parameters $\boldsymbol{\lambda} = (\lambda, k)$, whereby the integrals above can be generalized to

$$\int \mathrm{d}\lambda \rightarrow \int \mathrm{d}\boldsymbol{\lambda} = \sum_k \int \mathrm{d}\lambda \,. \tag{11}$$

After accounting for multiple species of quasiparticles, all the equations above can be applied to any integrable model. In fact, the only model-specific parameters that enter the calculations is the scattering phase, $\Theta(\lambda)$, encoding the interactions of the quasiparticles and the one-particle eigenvalues, $h_j(\lambda)$. These quantities can be obtained for a given model through the thermodynamic Bethe ansatz, and once plugged into the hydrodynamical equations the full framework of GHD can be applied to the problem.

The equations above constitutes the core of generalized hydrodynamics. Under evolution detailed by eq. (9) the system is always in a quasi-stationary state, from which various physical quantities can be calculated. In addition to solving eq. (9), iFluid also computes the so called characteristics [41] encoding the trajectories of the quasiparticles. Thus, the characteristics can be used for computing the hydrodynamics spreading of entanglement [19] and correlations [20]. The characteristics $U$ and $W$ have the simple interpretation as the inverse space and rapidity trajectories of the quasi-particles respectively [19], yielding

$$\vartheta(t, x, \lambda) = \vartheta(0, U(t, x, \lambda), W(t, x, \lambda)) \,. \tag{12}$$

Thus, $U(t, x, \lambda)$ is the position at time $t' = 0$ of the quasi-particle $\lambda$, which at time $t$ has the position $x$ [26]. Interestingly, the characteristics follow the same hydrodynamical equation as the filling function

$$\partial_t U(t, x, \lambda) + v^{\text{eff}}[\vartheta(t, x, \lambda)] \, \partial_x U(t, x, \lambda) + a^{\text{eff}}[\vartheta(t, x, \lambda)] \, \partial_\lambda U(t, x, \lambda) = 0 \,, \tag{13}$$

$$\partial_t W(t, x, \lambda) + v^{\text{eff}}[\vartheta(t, x, \lambda)] \partial_x W(t, x, \lambda) + a^{\text{eff}}[\vartheta(t, x, \lambda)] \, \partial_\lambda W(t, x, \lambda) = 0 \,, \tag{14}$$

with the initial conditions given by definition

$$U(0, x, \lambda) = x \,, \tag{15}$$

$$W(0, x, \lambda) = \lambda \,. \tag{16}$$

Hence, the characteristics can be propagated alongside the filling function with minimal numerical cost.

# 3 Core functionality of iFluid

iFluid implements all the equations listed in the section above along with several additional features. The main goal of the framework is to remain highly flexible while delivering fast performance. To achieve this goal, iFluid implements all its core methods in abstract classes, which must be extended by the user in order to supply the necessary model-specific functions required by the internal routines of iFluid. Several models and numerical solvers are already implemented in iFluid and will be demonstrated in Section 4. This section aims to introduce the iFluid base classes, while more in-depth information can be found in the documentation [47].

The hydrodynamical equations described in Section 2 are solved numerically by discretizing the integrals using appropriate quadratures. Thereby the linear integral equations are converted into linear matrix equations enabling very fast numerical calculation of the hydrodynamical quantities. One should note that iFluid employs a very strict convention for indices which is enforced through the `iFluidTensor` class. The discretized equations are found in Appendix B, while further information regarding the `iFluidTensor` is written in the documentation [47].

## 3.1 Implementing a model

A key feature of iFluid is its intuitive interface and extendibility. This is achieved through the abstract class `iFluidCore`, which implements all the equations of the thermodynamics Bethe ansatz from the previous section. Following the hydrodynamical principle the system is always in a quasi-stationary state, whereby all the methods of the `iFluidCore` can be applied for any given root density. However, in order to perform any specific calculations some explicit information regarding the model is required, namely the energy and momentum functions, $\epsilon(\lambda)$ and $p(\lambda)$, and the two-body scattering phase, $\Theta(\lambda - \lambda')$, along with their rapidity derivatives. Additionally, inhomogeneous systems require derivatives with respect to the couplings in order to compute eqs. 7 and 8. Hence, before any calculation can be undertaken one must extend the general `iFluidCore` class with a model-specific class `myModel < iFluidCore`, wherein the following abstract functions must be implemented

```
1  % Abstract methods which must be overloaded in model class
2  getBareEnergy(obj, t, x, rapid, type)
3  getBareMomentum(obj, t, x, rapid, type)
4  getEnergyRapidDeriv(obj, t, x, rapid, type)
5  getMomentumRapidDeriv(obj, t, x, rapid, type)
6  getScatteringRapidDeriv(obj, t, x, rapid1, rapid2, type1, type2)
7  getEnergyCoupDeriv(obj, coupIdx, t, x, rapid, type)
8  getMomentumCoupDeriv(obj, coupIdx, t, x, rapid, type)
9  getScatteringCoupDeriv(obj,coupIdx,t,x,rapid1,rapid2,type1,type2)
```

A few things to note about the input parameters: `t`, `x` and `rapid` are the spatial, temporal and rapidity arguments respectively. They can be either scalars or vectors. The `type` argument specifies the index of the quasiparticle type (only relevant for TBAs with multiple quasiparticle species). This argument can be either a scalar or an array of scalars and should default to 1 for single-particle TBAs. Lastly, the `coupIdx` input is a scalar index specifying which coupling the derivative is taken with respect to. The couplings are passed to the model-specific class through an array upon initialization (more on this later), whereby the array-index of the given coupling must match the `coupIdx`.

Although this might seem like a lot of work at first glance, most of these functions can be written in a single line. Examples of this can be found in the documentation [47].

## 3.2 Solving the GHD equation

Having specified the model-specific functions, all the equations listed in Section 2 except the main GHD equation (9) can be solved. Solving eq. (9) is achieved through the abstract iFluidSolver class. As previously mentioned, iFluid abstracts the algorithm for solving eq. (9) in order to accommodate future advances in the GHD numerics. Algorithms already implemented in iFluid can be found in the documentation [47], while new algorithms can be added simply by extending the iFluidSolver class and implementing the abstract methods

```
1  % Abstract methods which must be overloaded in sub—class
2  step(obj, theta_prev, u_prev, w_prev, t, dt)
3  initialize(obj, theta_init, u_init, w_init, t_array )
```

The method step() has the simple function of advancing the filling function by a single time step, $dt$, following eq. (9)

$$\vartheta(t, x, \lambda) \rightarrow \vartheta(t + dt, x, \lambda). \tag{17}$$

Several different approaches exists for taking this step. The solvers already implemented in iFluid utilize the implicit solution of eq. (9) [27, 34]

$$\vartheta\left(t', x, \lambda\right) = \vartheta(t, \tilde{x}(t', t), \tilde{\lambda}(t', t)), \tag{18}$$

where the functions $\tilde{x}(t', t)$ and $\tilde{\lambda}(t', t)$ are given by

$$\tilde{x}\left(t', t\right) = x - \int_t^{t'} d\tau \, v_\tau^{\text{eff}}(\tilde{x}(\tau, t), \tilde{\lambda}(\tau, t)) \tag{19}$$

and

$$\tilde{\lambda}\left(t', t\right) = \lambda - \int_t^{t'} d\tau \, a_\tau^{\text{eff}}(\tilde{x}(\tau, t), \tilde{\lambda}(\tau, t)). \tag{20}$$

The subscript of the effective velocity and acceleration denotes that the dressing is taken with respect to the filling function at time $\tau$. Further, note that the functions $\tilde{x}(t, 0)$ and $\tilde{\lambda}(t, 0)$ are in fact the characteristics $U(t, x, \lambda)$ and $W(t, x, \lambda)$ respectively. The step() function in iFluids FirstOrderSolver and SecondOrderSolver approximates solutions of eqs. (19) and (20) for a single time step at various orders.

Some algorithms require the filling function at only a single time to perform the step above, while others need the filling at multiple times in order to perform a more accurate step. For example, the class SecondORderSolver [27] employs a midpoint rule, whereby the midpoint filling function is stored within the class. However, in order to take the first step, the class needs to know the first midpoint, which is not given a priori. Thus, one must also implement the method initialize(), which prepares all the quantities necessary for starting the time evolution algorithm.

Once the abstracted methods above are implemented, one can solve eq. (9) simply by calling the method propagateTheta() within the iFluidSolver class. The method takes an initial state, $\vartheta(0, x, \lambda)$, and an array of time-steps, t array $= \{0, dt, 2\,dt, \dots, t_{\text{final}}\}$, as inputs and returns an array of filling functions

$$\texttt{theta\_array} = \{\vartheta(0, x, \lambda), \vartheta(dt, x, \lambda), \dots, \vartheta(t_{\text{final}}, x, \lambda)\}, \tag{21}$$

where each filling is stored as an iFluidTensor. In order to implement the abstract methods listed above, one will need some of the hydrodynamical equations listed in Section 2. Thus, the iFluidSolver constructor takes an iFluidCore object as argument and stores it. Whenever the dressing of a quantity (or something similar) is needed, one simply calls the appropriate method from the stored iFluidCore object.

# 4 Solving problems with iFluid

The previous section demonstrated how to implement models and algorithms in iFluid. Once the specific model and algorithm is implemented the hydrodynamical calculations become almost trivial, as most problems can be formulated in only a couple of lines of code. In the following examples we solve three distinctly different hydrodynamical problems using the already implemented methods of iFluid. The example codes can all be found on the iFLuid git page [48] and can be run in a matter of minutes on a laptop.

## 4.1 Quantum Newton's cradle

The original experimental realization of the quantum Newtons cradle [1] beautifully demonstrated integrability in a one-dimensional Bose gas. Recently, generalized hydrodynamics was utilized in a numerical study of the experiment [28], where the numerical results were obtained using the flea gas algorithm first described in [38]. Here we simulate a similar scenario of a Bose gas oscillating in a harmonic confinement. In contrast to previous studies which used an initial Bragg pulse to imprint different momenta unto the system, we simply displace half of the cloud with regards to the center of the trap akin to lifting a bead in the classical cradle. The initially displaced cloud will oscillate back and forth in the harmonic trap for several periods, thus demonstrating the lack of thermalization in integrable models. Furthermore, by keeping part of the cloud in the center we clearly illustrate the effect of the interparticle interaction, as the central cloud will be distorted upon overlapping with oscillating one.

The one-dimensional Bose gas is described by the Lieb-Liniger model with the Hamiltonian [30, 31]

$$\hat{H} = \int_0^L dx \left\{ \frac{1}{2m} \partial_x \hat{\psi}^\dagger(x) \partial_x \hat{\psi}(x) + c\hat{\psi}^\dagger(x)\hat{\psi}^\dagger(x)\hat{\psi}(x)\hat{\psi}(x) - \mu\hat{\psi}^\dagger(x)\hat{\psi}(x) \right\}. \qquad (22)$$

The interaction strength, $c > 0$, and the chemical potential, $\mu$, are the two couplings of the model, while $m$ is the particle mass and $L$ the system length. By employing an inhomogeneous chemical potential we can describe external traps through the local density approximation.

First we need to specify the problem at hand, namely the discretization grids, the couplings and the temperature. For this example we employ a rectangular quadrature for solving the integrals, whereby the quadrature weights are simply the grid spacings.

```
x_grid      = linspace(−6, 6, 2^7)
rapid_grid  = linspace(−13, 13, 2^7)     % linear grid
rapid_w     = rapid_grid(2) − rapid_grid(1)   % quad. weights
T           = 3 % temperature
```

Next, we have to specify the dynamical couplings used in the simulation. The couplings and their derivatives are declared as a cell array of anonymous functions with time and space arguments. The first row specifies the raw couplings, while the second and third row contains the temporal and spatial derivatives respectively. The class `LiebLinigerModel` requires the first column of the coupling array to be the chemical potential and the second column to be the interaction strength. The chemical potential is given by some central value, $\mu_0$, minus the harmonic confinement, while the interaction is simply set to unit:

```
5   sw          = @(t,x) 4.*x.^2      % single well potential
6   mu0         = 2                   % chemical potential offset
7
8   % specify couplings and their derivatives
9   couplings = { @(t,x) mu0 — sw(t,x) , @(t,x) 1 ; % (mu , c)
10               []                    , []         ; % d/dt
11               @(t,x) —8.*x          , []         } % d/dx
12
13  % initialize model
14  LL          = LiebLinigerModel(x_grid, rapid_grid, rapid_w, couplings);
```

Note that all operations in the anonymous functions should be elementwise (signified by the dot-prefix). Furthermore, entries for derivatives equal to zero can be left empty for a boost in performance.

Having specified the problem, we now turn to calculating the initial state given by two clearly separated, identical clouds. To illustrate the interaction between the two clouds, any distortion of the central cloud should be caused by the interactions. Hence, the initial state of the central cloud should be stationary with respect to the harmonic trap. This can be achieved in several fashions: Here we simply create an initial "double well", by displacing a copy of the harmonic trap via a heaviside function. In this case, the initial couplings are very different from the dynamical ones. Therefore, the method `calcThermalState()` of the `iFluidCore` class takes an initial set of couplings as optional argument, whereby:

```
15  % double—well potential with offset a
16  dw          = @(t,x,a) heaviside(—(x — a/2)).*sw(t,x) ...
17                     + heaviside( (x — a/2)).*sw(t,x—a)
18
19  % specify initial couplings (no deriv needed)
20  offset      = 3
21  coup_init  = { @(t,x) 2 — dw(t,x,offset) , @(t,x) 1 }
22
23  % calculate thermal state
24  theta_init = LL.calcThermalState(T, coup_init);
```

Finally, we are ready to solve the GHD equation (9), using the `SecondOrderSolver` class [27]. Simply pass the model to the solver and run the simulation through the `propagateTheta()` method:

```
25  Solver2 = SecondOrderSolver(LL) % initialize solver
26  t_array = linspace(0, 8, 321) % dt = 0.025
27  theta_t = Solver2.propagateTheta(theta_init, t_array)
```

The final output is a cell array, where the $i$'th entry is the filling function, $\vartheta(x, \lambda)$, at the time `t_array(i)`. Note, each $\vartheta(x, \lambda)$ is stored as an `iFluidTensor`.

The 27 lines of code above is all there is needed for specifying and solving a typical problem in iFluid. According to the hydrodynamic principles, the system is in a quasi-stationary state at every point in time. Hence, once the filling function is computed, it can be used for any calculation within the thermodynamic Bethe ansatz.

We start out by illustrating the motion of the two clouds of Bose gases in the Newton's cradle by calculating the linear (atomic) density corresponding to the zeroth charge density in eq. (1). Given the filling function, we can also calculate the root density, $\rho(t, x, \lambda)$, thus illustrating the motion of the quasiparticles.

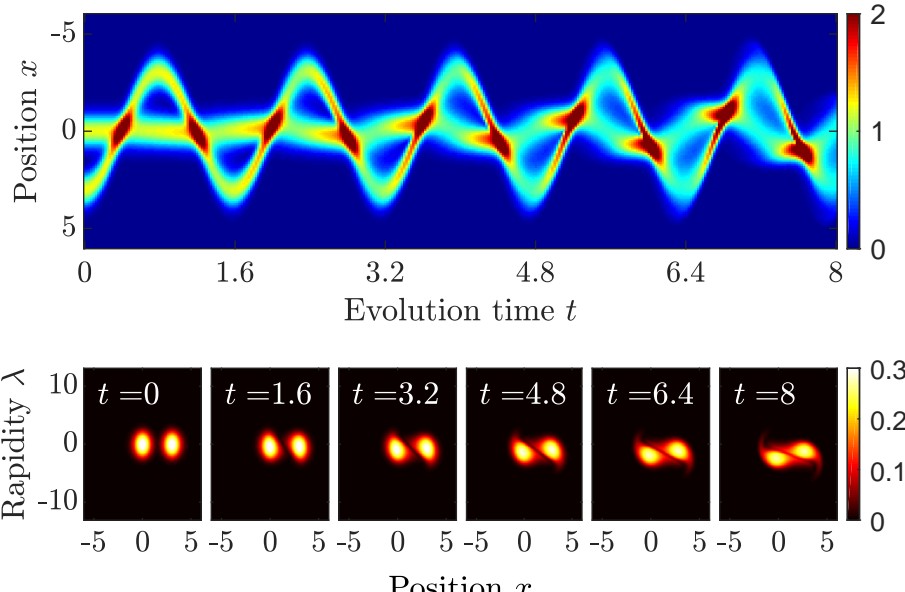

Figure 1: Above: Evolution of the linear density of a Bose gas in a Newton's cradle. The interactions between the oscillating and stationary cloud transfers momentum between them. At the end of the evolution, the system moves as a single cloud exhibiting both a center-of-mass and breathing motion. Below: Snapshots of the root density at each period. Initially the two clouds have distinct root densities, which gradually merges into a single, binary distribution.

```
28   charge_idx = 0 % linear density is 0th charge density
29   n_t = LL.calcCharges(charge_idx, theta_t, t_array)
30   rho_t = LL.transform2rho(theta_t, t_array)
```

The evolution of the linear density is plotted in Figure 1 along with the root density at selected times. In the root density picture we initially see two blobs well separated in space and centred on zero rapidity. The central blob is the stationary state of the harmonic trap and thus remains in place, while the offset blob is accelerated by the harmonic confinement. This causes the offset root density to encircle the central one, effectively resulting in an oscillating motion of the density. Every time the two clouds overlap interactions occur, effectively transferring a small portion of the oscillating clouds momentum to the central one. Their total interaction is primarily determined by two things: the interaction strength, $c$, and the amount of time at which the clouds overlap. By separating the clouds by only a small distance, the offset cloud will only accelerate to a low velocity before passing the stationary cloud. Thus, the overlap time becomes long leading to a large distortion of the root densities. Therefore, the two blobs partially merge after merely a few oscillations, creating a binary system orbiting the center while rotating around itself. In the density picture this produces a single cloud whose center of mass oscillates in the harmonic trap while the cloud itself exerts a breathing motion.

Figure 1 clearly demonstrates the collective interactions in generalized hydrodynamics. In a non-interacting theory, the offset cloud would simply have encircled the center forever without any deformations of the root densities. Meanwhile, any non-integrable system would have rapidly thermalized producing a single, Gaussian cloud.

Next, we wish to calculate the characteristic, $U$ and $W$ of eqs. (12), and illustrate their interpretation. This is easier if the two blobs of quasiparticles stay separate, which can be achieved simply by starting them further apart, thus decreasing their effective interaction over

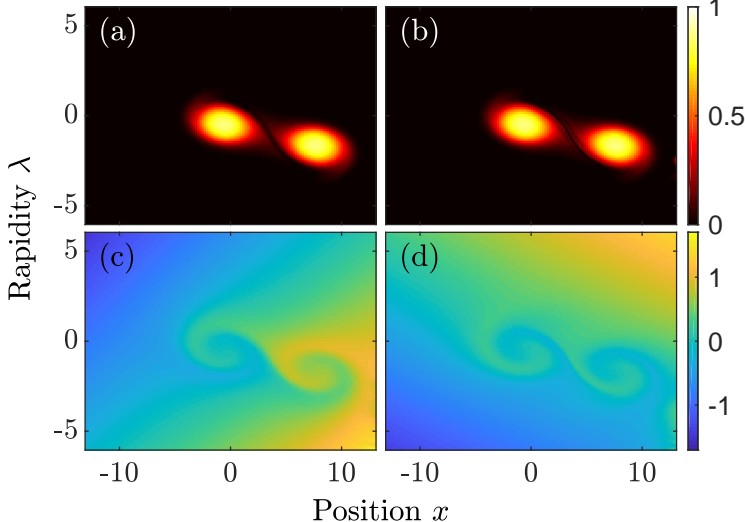

Figure 2: Demonstration of the solution by characteristics in eq. (12). **(a)** Filling function after five periods $\vartheta(t_{\text{final}}, x, \lambda)$. **(b)** Initial filling function interpolated to characteristics $\vartheta(0, U(t_{\text{final}}, x, \lambda), W(t_{\text{final}}, x, \lambda))$. This indeed reproduces the filling shown in **(a)**. The characteristics $U$ and $W$ rescaled by their initial max value are depicted in **(c-d)** respectively. The spiral patterns are a result of the interaction between the clouds.

time. The characteristics follow the same hydrodynamical equations as the filling function, and can be computed alongside the filling:

```
31  [theta_t, U_t, W_t] = Solver2.propagateTheta(theta_init, t_array)
```

Just like the filling, the characteristics are returned as cell arrays of `iFluidTensor`. According to eq. 12, the filling function after some evolution time can be inferred from initial state via the characteristics. To demonstrate this we interpolate the initial filling to the characteristics at some time $t$ and compare with $\vartheta(t)$. Performing the interpolation is straightforward in Matlab:

```
32  % Get the distribution of the first (and only) quasiparticle type. Return as a
        matrix of double.
33  theta     = theta_t{end}.getType(1, 'double')
34  U         = U_t{end}.getType(1, 'double')
35  W         = W_t{end}.getType(1, 'double')
36
37  % interpolate theta_init to (U(t_final) , W(t_final))
38  theta_UW = interp2( x_grid, rapid_grid, plt(theta_init), U(:), W(:) )
39  theta_UW = reshape( theta_UW, length(rapid_grid), length(x_grid) )
```

The resulting filling function is plotted in Figure 2 along with the characteristics. Starting with the characteristics we observe a spiral pattern, which provides new information about the quasiparticle trajectories not accessible from the root densities themselves. Although the central blob is stationary with respect to the harmonic confinement, its quasiparticles still have a finite velocities causing them to move in an orbit around the center of the blob. In fact, in the case of a harmonic confinement and no interactions, all the quasiparticles move in closed orbit around $(x, \lambda) = (0, 0)$. However, the interactions between the two clouds distort the quasiparticle trajectories resulting in the observed spiral patterns. Interpolating the initial filling to the characteristics does indeed reproduce the filling function at time $t$ as seen in Figures 2. The small differences between the two fillings are due to inaccuracies in

**Sci|Post**                                                    SciPost Phys. 8, 041 (2020)

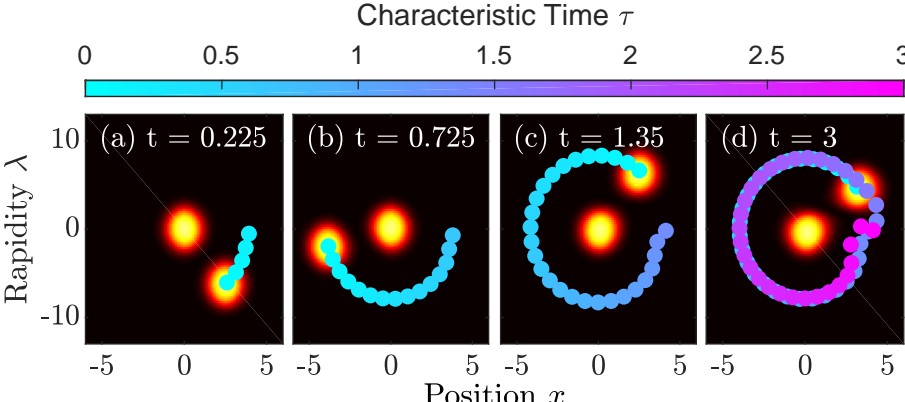

Figure 3: Interpretation of characteristics as inverse quasiparticle trajectories. The plots show the filling function at various times, $\vartheta(t)$, overlaid with the characteristic coordinates $(U(\tau), W(\tau))$. The coordinates mark the location of the quasiparticle at time $t - \tau$, which later at time $t$ is found in the center of the encircling filling. The clouds interact whenever they overlap in space causing a distortion of the quasiparticle trajectories. For detailed description refer to main text.

the interpolation and the finite number of grid points, making it very hard to resolve the fine structure of the characteristics.

The interpretation of the characteristics can be further understood by plotting them as function of time. Going back to definition in Section 2, we recollect that the characteristics encode the positions and rapidities of the quasiparticles at an earlier time. Note that this is *not* the trajectory of the quasiparticle but rather the inverse trajectory. Figure 3 depicts the filling at different times along with characteristics of the quasiparticles in the center of the offset blob as a function of time. Starting with Figure 3.(a), the bullets mark represent the coordinates at time $t - \tau$ of the quasiparticles now located in the center of the offset blob. The characteristics depict a circular motion, as the two blobs have yet to overlap. However, as the two cloud pass through each other, the quasiparticle trajectories become distorted, as seen in Figure 3.(b). This becomes especially clear when looking at the point $t = \tau$, which clearly has moved. Hence, due to the distortion of the trajectories, a different quasiparticle of the initial root density can now be found in the center of the blob. Every time the two clouds overlap the trajectories become increasingly distorted, which in the end produced the spiral patterns observed previously in Figure 2. Thus, the characteristics do indeed encode the original location of the quasiparticles, which can be used to infer correlations of entanglements of the system [19, 20].

## 4.2   Charges and currents of XXZ model

The Heisenberg XXZ model is another integrable model already implemented in iFluid. It has the Hamiltonian [42]

$$\hat{H} = \sum_{j=1}^{N} \left( \hat{S}_j^x \hat{S}_{j+1}^x + \hat{S}_j^y \hat{S}_{j+1}^y + \Delta_j \hat{S}_j^z \hat{S}_{j+1}^z + B_j \hat{S}_j^z \right), \tag{23}$$

where $\hat{S}_j^\sigma$ are the standard spin-$\frac{1}{2}$ operators, while the couplings are the magnetic field, $B$, and the interaction, $\Delta$. The model differs from the Lieb-Liniger model in a number of different ways. In the case of $\Delta \geq 1$ the quasiparticles are restricted to the first Brillouin zone in the rapidity. Furthermore, bound states within the chain can occur, which requires a TBA

description consisting of multiple root densities, i.e. multiple types of quasiparticles otherwise known as strings. For $B < 0$ infinitely many root densities are needed in theory, however, in practice one can truncate this to a relatively small number, as each additional root density has diminishing effect.

In this example we examine a system initially confined by a strong magnetic field, which afterwards is gradually decreased. Setting up the problem in iFluid is very similar to the previous example. Since the expression for the coupling is a little longer in this case, we use Matlab's symbolic feature to take the derivative of the coupling for us. Furthermore, we solve the TBA integrals using a Legendre-Gauss quadrature obtained via the `legzo()` method [49].

```matlab
1  Ntypes = 3    % number of quasiparticle types
2  dt     = 0.01 % time step
3  tmax   = 1    % t_final
4  T      = 1    % temperature
5
6  % Rapidity is confined within first Brillouin zone
7  % To solve integrals we employ a Legendre—Gauss quadrature
8  [rapid_grid,rapid_w] = legzo(2^7, —pi/2, pi/2)
9  x_grid  = linspace(—1.5, 1.5, 2^7)
10 t_array = linspace(0, tmax, tmax/dt+1)
11
12 % Define couplings via Matlab symbolic
13 syms x t
14
15 B      = —1 — (1—tanh(3*t))*10*x.^2 % symbolic function
16 B_func = matlabFunction( B )         % anonymous function
17 dBdt   = matlabFunction( diff(B,t) )
18 dBdx   = matlabFunction( diff(B,x) )
19
20 couplings = { B_func , @(t,x) acosh(1.5) ; % (B , acosh(Delta))
21             dBdt   , []                 ; % d/dt
22             dBdx   , []                 } % d/dx
23
24 % Instantiate model and solver and solve the GHD eq.
25 XXZ     = XXZchainModel(x_grid, rapid_grid, rapid_w, couplings, Ntypes)
26 Solver2 = SecondOrderSolver(XXZ)
27
28 theta_0 = XXZ.calcThermalState(T)
29 theta_t = Solver2.propagateTheta(theta_0, t_array)
30
31 % Set B to 0 and calculate exp. values of charges + currents
32 XXZ.setCouplings( {@(t,x) 0 , @(t,x) acosh(1.5)} )
33 [q_t, j_t] = XXZ.calcCharges([0 2], theta_t, t_array)
```

In the final two lines of the code we calculate the expectation values of the zeroth and second charges and associated currents. We wish to calculate the kinetic energy, thus the energy without the contribution from the magnetic field. Hence, we simply set the field to zero before the calculation.

Figure 4 shows the calculated quantities at select times. Starting with subfigure 4.(a) we see the evolution of the linear density. Initially the density profile consists of a single, smooth curve dictated by the parabolic, confining magnetic field. However, the temporal change in the coupling induces force terms on the quasiparticles, and each string experiences a different effective acceleration. Thus, the strings starts moving outwards at different velocities, whereby three distinct density profiles become visible after some time. These three profiles correspond

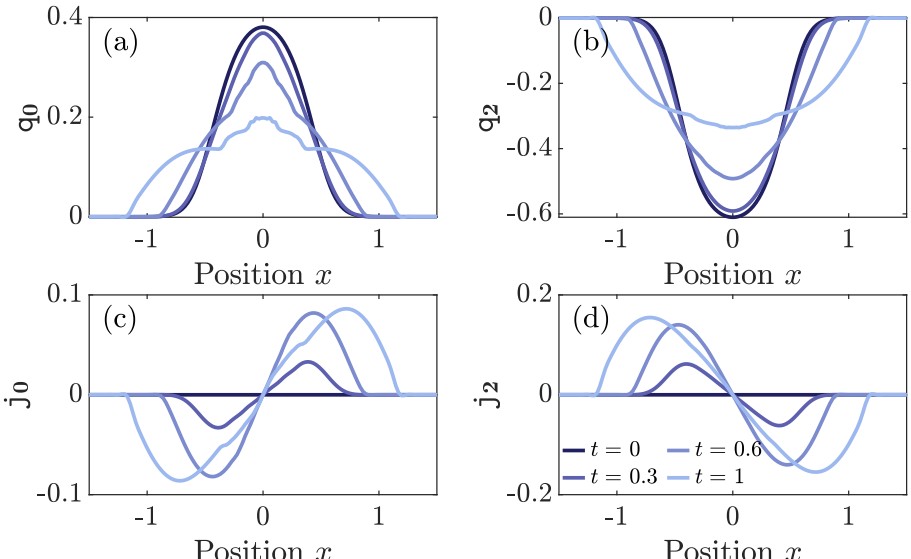

Figure 4: Expectation values of charge densities and their associated currents as function of $x$ for the XXZ model. The system is initialized in a parabolic, confining magnetic field, whereafter the field is gradually lowered. **(a,c)** Linear density and its current at different times. As the field is lowered the difference in velocity between the different quasiparticles becomes apparent. **(b,d)** Kinetic energy density and current at different times. The higher kinetic energy of first order quasiparticles hides the contribution from higher orders.

to the three root densities accounted for in the calculation. This further emphasises the point made earlier that each additional root density included has diminishing effect. The same three-part structure can be seen in the associated current, albeit to a much lesser degree. Meanwhile, the expectation value of the kinetic energy barely changes by taking the contribution from additional strings into account, as the string corresponding to no bound states has the largest kinetic energy.

## 4.3 Partitioning protocol in relativistic sinh-Gordon

In our final example we examine the relativistic sinh-Gordon model with the Hamiltonian [50, 51]

$$\hat{H} = \int dx \left\{ \frac{c^2}{2} \pi^2(x) + \frac{1}{2} [\partial_x \phi(x)]^2 + \frac{\beta^2 c^2}{g^2} : \cosh[g \phi(x)] : \right\} , \qquad (24)$$

where the constant $c$ is the speed of light, and $: \bullet :$ denotes normal ordering with respect to the ground state. The mass-parameter $\beta$ is related to the physical renormalized mass $m$ by

$$m^2 = \beta^2 \frac{\sin(\alpha \pi)}{\alpha \pi} \quad \text{and} \quad \alpha = \frac{c g^2}{8\pi + c g^2} . \qquad (25)$$

In the iFluid implementation of the model, the couplings are given by the renormalized interaction, $\alpha$, and the parameter $\beta$. Additionally, iFluid adds a chemical potential to the Hamiltonian as a third coupling, such that users can control the initial linear density.

With this example we wish to illustrate how iFluid deals with extensive systems. Thus, we explore a well-known protocol in the GHD community [14, 15], namely a partitioning protocol where two semi-infinite, homogeneous systems, or leads, of different temperature

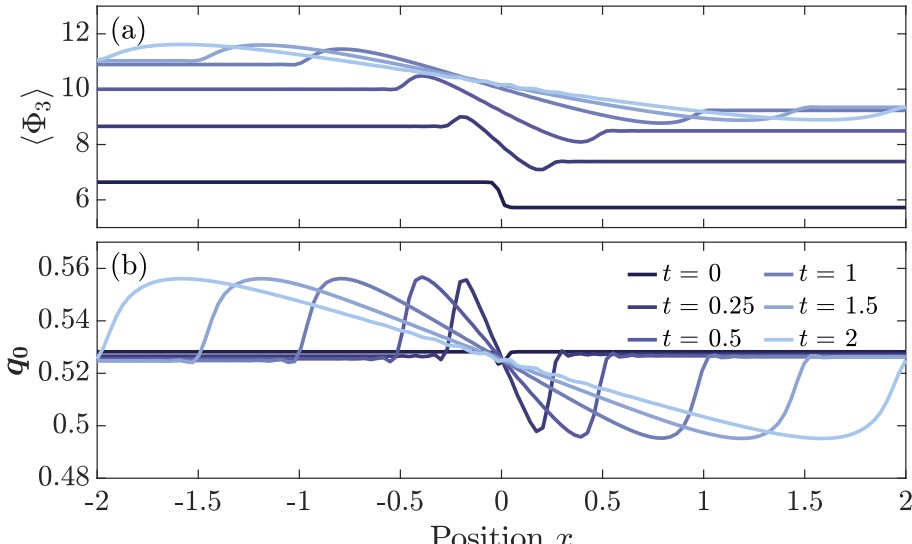

Figure 5: Partitioning protocol in relativistic sinh-Gordon model, where two homogeneous, semi-infinite leads of temperature $T_L = 1.25$ and $T_R = 1.75$ are joined together at time $t = 0$. Additionally the interaction, $\alpha$, is gradually increased. **(a)** Expectation of the vertex operator $\Phi_k(x) \equiv\, : e^{kg\phi(x)} :$ for $k = 3$ at different times. As the interaction is increased globally, so is $\langle \Phi_3(x) \rangle$. **(b)** Evolution of the linear density driven by the difference in temperature. Quasiparticles from the hot reservoir travel quickly into the cold reservoir creating an expanding density wave.

are joined together. Since iFluid uses finite-sized grids, we obviously cannot store an infinitely long system. However, we can toggle the option `extrapFlag = true` of the `iFluidSolver` to enable extrapolation of the filling functions upon propagation. Usually extrapolation is ill advised, but in this case the extrapolation works well since each lead is homogeneous.

There are several ways of realizing the two leads. One option is declaring a space-dependent temperature and then balancing the difference in density via the chemical potential. Achieving exactly equal density of the two leads can be tricky using this approach, however, the example here is merely a demonstration of using the model. In addition to performing the regular partitioning protocol we also gradually increase the interactions, such that $\alpha(t) = (1 + 0.5 \tanh(2t))/(8\pi + 2)$, while $\beta = 1$. By now the process of setting up grids and couplings in iFluid should be known to the reader, so we merely show the part of the code unique to the problem:

```
1   T          = @(x) 1.5 + 0.25*tanh(50*x) % inhomogeneous temperature
2
3   options.extrapFlag = true % options struct for solver
4
5   shG        = sinhGordonModel(x_grid, rapid_grid, rapid_w, couplings)
6   Solver2    = SecondOrderSolver(shG, options)
7
8   theta_init = shG.calcThermalState(T, coup_balance)
9   theta_t    = Solver2.propagateTheta(theta_init, t_array)
10
11  % calculate expectation value of density and vertex operators
12  q0_t       = shG.calcCharges(0, theta_t, t_array)
13  Psi3_t     = shG.calcVertexExpval( 3, theta_t, t_array)
```

At the end of the calculation the linear density is calculated along with the expectation value of the vertex operator [51], $\Phi_k(x) \equiv\, : e^{kg\phi(x)} :$, which are both plotted in Figure 5.

Starting with the linear density we confirm the density being homogeneous initially. However, the higher temperature of the right lead causes a larger number of quasiparticles to be initialized at higher rapidities, whereby they move at higher velocity. Thus, quasiparticles from the right travel into the left lead faster than the quasiparticles from the left can fill out the void left behind. Therefore, one observes a wave of increased density travelling left and another wave of decreased density travelling right.

The change in interaction, $\alpha$, has barely any influence on the redistribution of density. However, it greatly influences the expectation values of the vertex operators, since the expression for $\langle \Phi_k \rangle(x)$ is directly dependent on $\alpha$ [51]. Thus, as the interaction increases, so does $\langle \Phi_k \rangle(x)$. Even the regions not yet reached by the density wave are affected, since $\alpha$ is changed globally.

## 5   Conclusion

We have demonstrated that iFluid enables the user to perform state of the art GHD calculations in only a few lines of code. Additionally we have shown that iFluid can be easily extended to encompass a large number of integrable models and numerical solvers. We hope that iFluid will be a help to both students and researchers, who wish to explore the numerical side of generalized hydrodynamics. Furthermore, the recent experimental evidence of GHD's ability to describe the dynamics of cold gas experiments [29] further increases the need for powerful numerical tools. Thus, iFluid represents a great step forward in making the theory more widely accessible, since no open-source software exist in the GHD community so far.

Aside from being easy to use, iFluid also offers great extendibility through its abstract classes. Thus, users can implement new models simply by extending the `iFluidCore` class and overloading a couple of methods. Similarly, new solvers of the GHD Euler-equation (9) can be added to the framework by extending the class `iFluidSolver`. Well-established algorithms from the field of fluid dynamics can thereby be seamlessly added and tested.

The development of iFluid is an ongoing process, as more and more advances are being made in the theory of generalized hydrodynamics. By employing a modular layout, the framework aims to function as a fundamental platform on which further tools can be built upon. Applications for calculating the hydrodynamical spreading of correlations and entanglement seem especially promising.

The current version of iFluid is written is Matlab, however, plans are currently in the works to write the framework as either a Python package or a C++ library. While each language has its own advantages, the Matlab iteration of iFluid is easily accessible to most members of the GHD community while retaining decent performance. We also welcome anyone interested in contributing to the project to contact the authors through either email or the official iFluid repository on Github.

## Acknowledgements

The authors would like to thank Alvise Bastianello, Bruno Bertini, and Vincenzo Alba for enlightening discussions on the topic of generalized hydrodynamics. We also thank Mohammadamin Tajik, Federica Cataldini and João Sabino for proofreading the manuscript. Finally, we would like to thank Sebastian Erne and Camille Leveque for discussions and inputs on the code.

**Funding information**   F.M. acknowledges the support of the Doctoral Program CoQuS. This research was supported by the SFB 1225 'ISOQUANT' and grant number I3010-N27, financed by the Austrian Science Fund (FWF), and the Wiener Wissenschafts- und TechnologieFonds (WWTF) project No MA16-066 (SEQUEX).

# A   Thermodynamic Bethe ansatz of implemented models

The thermodynamic Bethe ansatz is a textbook technique nowadays [42], which can be applied to a large range of integrable models. In this section we report the basic TBA quantities of the models highlighted in the main text and emphasise the details of the iFluid implementation.

## A.1   Lieb-Liniger model

The Lieb-Linger model describes a one-dimensional Bose gas with contact interactions governed by the Hamiltonian [30, 31]

$$
\hat{H} = \int_0^L dx \left\{ \frac{1}{2m} \partial_x \hat{\psi}^\dagger(x) \partial_x \hat{\psi}(x) + c \hat{\psi}^\dagger(x) \hat{\psi}^\dagger(x) \hat{\psi}(x) \hat{\psi}(x) - \mu \hat{\psi}^\dagger(x) \hat{\psi}(x) \right\} , \tag{26}
$$

where $\hat{\psi}^\dagger(x), \hat{\psi}(x)$ are the bosonic fields, while $c$ is the interaction strength and $\mu$ is the chemical potential. The TBA detailed here and implemented in iFluid is only valid for repulsive interactions $c > 0$. Thus, the three main functions (single-particle energy, momentum and scattering) required for solving the GHD equations read

$$
\epsilon(\lambda) = \frac{\lambda^2}{2m} - \mu \quad , \quad p(\lambda) = \lambda \quad , \quad \Theta(\lambda) = \arctan\left( \frac{4\lambda mc}{\lambda^2 - (2mc)^2} \right) . \tag{27}
$$

The iFluid implementation of the Lieb-Liniger model is contained in the `LiebLinigerModel` class, which takes the chemical potential as the first coupling and the interaction strength as the second coupling. In addition to the standard TBA equations, the `LiebLinigerModel` class implements additional methods:

```
1  % Given external potential V_ext, fit mu to get given number of atoms
2  fitAtomnumber(obj, T, V_ext, Natoms, mu0_guess, setCouplingFlag)
3
4  % Calculate the n—body local correlator g_n
5  calcLocalCorrelator(obj, n, theta, t_array)
```

The first method `fitAtomnumber()` fits the central chemical potential, $\mu_0$, where $\mu = \mu_0 - V_{ext}(x)$, to achieve a thermal state whose root density integrates to a specified number of atoms. This is especially useful for experimental comparisons. The second method `calcLocalCorrelator()` computes the local n-body correlator through the approach detailed in [21].

Recently generalized hydrodynamics was shown to describe the dynamics of an experimentally realized one-dimensional Bose gas. Hence, the theory appears to be a powerful tool for simulating real systems. Therefore, iFluid implements a wrapper class `LiebLinigerModel_SI` for converting inputs in SI-units to the internal units of iFluid, which then calls the appropriate methods of the `LiebLinigerModel` class.

For more detailed description of these methods we refer the reader to the official iFluid documentation [47].

## A.2 XXZ spin chain model

The XXZ spin chain model is a discrete integrable model of $N$ sites with the Hamiltonian

$$\hat{H} = \sum_{j=1}^{N} \left( \hat{S}_j^x \hat{S}_{j+1}^x + \hat{S}_j^y \hat{S}_{j+1}^y + \Delta_j \hat{S}_j^z \hat{S}_{j+1}^z + B_j \hat{S}_j^z \right) . \tag{28}$$

Here the standard spin-$\frac{1}{2}$ operators are $\hat{S}_j^\sigma$. while $B_j$ denotes the magnetic field at site $j$ and $\Delta_j$ the interaction. Although the model is discrete, it is treated exactly like the continuous models in the hydrodynamical description.

As already mentioned in the main text, the TBA description of the XXZ chain requires multiple root densities. The exact thermodynamics of the model are highly dependent on the values of $B$ and $\Delta$ [42]. The iFluid implementation of the model is only valid in the sector of $B < 0$ and $\Delta \geq 1$, in which an infinite set of root densities, $\{\rho_k(\lambda)\}_{k=1}^{\infty}$, are required for the TBA description. The rapidities of every root density are confined to the first Brillouin zone $\lambda \in [-\pi/2, \pi/2]$, and their associated functions read [27]

$$\epsilon_k(\lambda) = -\frac{1}{2} \sinh(\theta) \partial_\lambda p_k(\lambda) - kB \quad , \quad p_k(\lambda) = 2 \arctan\left[ \coth\left( \frac{k\theta}{2} \right) \tan\lambda \right] , \tag{29}$$

with the scattering

$$\Theta_{k,l}(\lambda) = \left( 1 - \delta_{k,l} \right) p_{|k-l|}(\lambda) + p_{k+l}(\lambda) + \sum_{n=1}^{\min(k,l)-1} 2p_{|k-l|+2n}(\lambda) . \tag{30}$$

The iFluid implementation of the model, XXZchainModel, takes $B$ as the first coupling, while the angle $\theta = \operatorname{arccosh} \Delta$ serves as the second coupling.

## A.3 Relativistic sinh-Gordon model

The sinh-Gordon model is a relativistic field theory described by the Hamiltonian [50, 51]

$$\hat{H} = \int dx \left\{ \frac{c^2}{2} \pi^2(x) + \frac{1}{2} [\partial_x \phi(x)]^2 + \frac{\beta^2 c^2}{g^2} : \cosh[g\phi(x)] : \right\} , \tag{31}$$

where

$$m^2 = \beta^2 \frac{\sin(\alpha\pi)}{\alpha\pi} \quad \text{and} \quad \alpha = \frac{cg^2}{8\pi + cg^2} . \tag{32}$$

Like the Lieb-Liniger model, the thermodynamic Bethe ansatz is determined by only a single root density. The TBA functions of the model read

$$\epsilon(\lambda) = m \cosh\lambda - \mu \quad , \quad p(\lambda) = m \sinh\lambda \quad , \quad \Theta(\lambda) = i \log \frac{\sinh\lambda - i\sin(\alpha\pi)}{\sinh\lambda + i\sin(\alpha\pi)} , \tag{33}$$

where we have added a chemical potential, $\mu$, to the energy function.

The iFluid implementation of the model, sinhGordonModel, takes $\alpha$ as the first coupling, $\beta$ as the second coupling, and $\mu$ as the third coupling. In the addition to the standard TBA functions, the sinhGordonModel class also implements methods for calculating the expectation values of vertex operators using the approach detailed in [51].

```
1  % Calculates <Psi_k> up to order k_max
2  calcVertexExpval( obj, kmax, theta_t, t_array )
```

# B   Numerical implementation of GHD equations

In this section we summarize iFluids numerical implementation of the equations in Section 2. All the equations listed in this section are implemented as separate functions in the `iFluidCore` class. Once again we refer to the official documentation for more in-depth information regarding the input and output quantities of these functions. The most challenging task in solving these equations is keeping track of all the indices, which iFluid handles through the `iFluidTensor` class.

## B.1   Tensor representation and index conventions

The main quantity of TBA is the root density (or filling function). The root density is explicitly dependent on the rapidity, $\lambda$, but can also have a spatial dependence by assuming the root density at neighbouring points to be slightly different (known as the local density approximation). Finally, some TBAs (like the XXZ model) require multiple root densities. Thus, the thermodynamics of the state is fully determined by the set $\{\rho_k(x,\lambda)\}_{k=1}^{N_{\text{types}}}$. By discretizing space and rapidity all the information contained within this set can be stored in a single rank-3 tensor, where each of its indices corresponding to the dependencies listed above. Most quantities in the GHD framework abide to a similar structure except for the scattering phase, $\Theta$, which details the scattering between quasiparticles of different rapidities and types. Thus, the discretized scattering phase is a rank-5 tensor, as it requires an additional rapidity and type argument. Keeping track of all these indices can be very tedious. Therefore, iFluid implements the `iFluidTensor` class, which overloads many of Matlabs standard matrix operations in order to handle the extra indices. The `iFluidTensor` assumes the following order of the indices:

```
1  % iFluidTensor index convention
2  index 1: main rapidity index
3  index 2: spatial index
4  index 3: main type index
5  index 4: secondary rapidity index used in convolutions
6  index 5: secondary type index used in convolutions
```

Note how the `iFluidTensor` does not a have temporal index despite GHD providing solutions to the dynamics. Instead, each `iFluidTensor` represents the quasi-stationary state of the system at some point in time. Methods like `propagateTheta()` of the `iFluidSolver` class thus return a cell array, where each entry is an `iFluidTensor` representing the system at a given time.

In all the examples shown in Section 4 no direct manipulation of `iFluidTensors` was needed, however, the class is a critical data structure for extending the iFluid framework and adding more modules. More information regarding the `iFluidTensor` class can be found in the official documentation.

## B.2   Discretized GHD equations

The integral equations of Section 2 are solved by appropriate quadratures, whereby the equations transform into matrix equations. Following the strict index convention of iFluid, we distinguish the indices by writing the rapidity index as subscript and the type index as superscript. Like the equations in the main text, the spatial argument is omitted for readability. However, in the discretized form the spatial argument just enters as an additional index where

each entry is multiplied elementwise. Thus, the integrals transform as follows

$$\int d\lambda\, h(\lambda) \rightarrow \sum_i \sum_k w_i^k h_i^k \,, \tag{34}$$

$$\int d\lambda'\, \Theta(\lambda, \lambda')h(\lambda') \rightarrow \sum_i \sum_k w_i^k \Theta_{ji}^{lk} h_i^k \,, \tag{35}$$

where $w_i^k$ are the quadrature weights.

Employing all the conventions above, we can express the expectation values of charge densities (1) and associated currents (2) as

$$\mathsf{q}_n = \sum_i \sum_k w_i^k \rho_i^k (h_n)_i^k \tag{36}$$

and

$$\mathsf{j}_n = \sum_i \sum_k w_i^k \rho_i^k (h_n)_i^k (v^{\text{eff}})_i^k \,. \tag{37}$$

In order to compute the velocity, we need to solve the dressing equation (4). In its discretized form the equation reads

$$(h^{\text{dr}})_i^k = h_i^k - \sum_j \sum_l w_j^l T_{ij}^{kl} \vartheta_j^l (h^{\text{dr}})_j^l \,, \tag{38}$$

where we have written $T_{ij}^{kl} = (\partial_\lambda \Theta)_{ij}^{kl}/2\pi$ for more compact notation. The dressing of quantities constitutes the greatest operational cost of most of the GHD algorithms. One can rewrite the dressing equations into a system of linear equations, which can be solved by the standard routines of Matlab. Thus, the dressing is achieved by solving

$$h_i^k = \sum_j \sum_l \left( \delta_{ij} \delta^{kl} + w_j^l T_{ij}^{kl} \vartheta_j^l \right)(h^{\text{dr}})_j^l \,, \tag{39}$$

where $\delta_{ij}$ is the Kronecker delta. Finally, the force terms (7) and (8) of the effective acceleration are computed by solving

$$f_i^k = -(\partial_\alpha p)_i^k + \sum_j \sum_l w_j^l T_{ij}^{kl} \vartheta_j^l ((\partial_\lambda p)^{\text{dr}})_j^l \tag{40}$$

and

$$\Lambda_i^k = -(\partial_\alpha \epsilon)_i^k + \sum_j \sum_l w_j^l T_{ij}^{kl} \vartheta_j^l ((\partial_\lambda \epsilon)^{\text{dr}})_j^l \,. \tag{41}$$

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
