# Peer review of "Introducing iFluid: a numerical framework for solving hydrodynamical equations in integrable models"

_SciPost Physics, doi:SciPost Phys. 8, 041 (2020)_

## Round 1 · Referee Report · Anonymous · 2020-3-3

Strengths

1- First open source GHD (generalized hydrodynamics) code
2- More models can be added easily
3- Appears to be relatively easy to use

Weaknesses

1- Matlab (proprietary language) is an unfortunate choice for an open-source project. Python or Julia would have been much more appropriate.
2- More methods for solving GHD equations could have been included, such as the flea gas.

Report

In this paper, the authors introduce an open-source Matlab code to simulate the non-equilibrium dynamics of integrable systems using the recently introduced framework of generalized hydrodynamics (GHD). The current version of the code can solve the dynamics of the Lieb-Liniger model, XXZ spin chain in the gapped regime, and of the sinh-Gordon model; in arbitrary (but smooth) time and position-dependent potential.

I believe this is a valuable contribution as it should make GHD a lot more accessible. The code seems to be easy to use and I expect it will greatly benefit the community. The code is modular and other integrable models or other solvers can be implemented. The paper is well-written and easy to read, and the examples nicely illustrate the functionalities of the code. It is unfortunate that the authors chose Matlab -- a proprietary programming language -- for this open source project. I'm hoping to see a C++, Julia or Python version in the future.

Requested changes

I am recommending publication as is.

---

## Round 1 · Referee Report · Anonymous · 2020-3-5

Strengths

- This work is timely as the dynamics of integrable systems is a topic of intensive experimental study, and this package automates a lot of GHD, thus making it easy for other experimental groups to compare their results to GHD.

- The introduction of spacetime inhomogeneities is a great strength as these are usually experimentally relevant.

- The introduction to GHD is admirably clear.

- For what it's worth I do not entirely agree with the other referee about MATLAB. Especially given the prospect that this work will be widely adopted by experimental groups, I think the choice of language is appropriate.

Weaknesses

I do not see any major weaknesses (though as the other referee points out there is room for extension in various directions).

Report

I recommend publication as is.

---

## Editorial Decision

published